# Discussion on Adjustment Method of the Characteristic Period of Site Response Spectrum with Soft Soil Layer

Yuandong Li, Bing Hao, Zhen Chen, Zhenghua Zhou *, Zhu Bian, Yi Han and Cheng Peng

College of Transportation Engineering, Nanjing Tech University, Nanjing 210009, China;
wddd2017@njtech.edu.cn (Y.L.)
* Correspondence: bjsmoc@njtech.edu.cn

**Abstract:** Twelve site models were established based on the analysis of the influence of site conditions on earthquake damage and the influence of the soft soil layer on-site seismic response. The equivalent linearization site seismic response analysis is carried out at different input ground motion levels to discuss the influence of soft soil layer thickness and buried depth. The results show that the characteristic period of the response spectrum exhibits a gradual increase as the buried depth or thickness of the soft soil layer increases. Furthermore, the characteristic period of the response spectrum also increases with the rise in the input ground motion peak. Moreover, according to the influence characteristics of soft soil thickness, buried depth, and input ground motion intensity on the characteristic period of the site acceleration response spectrum, a method for adjusting the characteristic period of the site acceleration response spectrum with a soft soil layer is put forward.

**Keywords:** site conditions; ground motion; characteristic period; equivalent linearization; earthquake damage

## 1. Introduction

Site conditions play a decisive role in the influence of ground motion [1,2]. In engineering seismic design, the engineering design and construction solutions that are compatible with the characteristics of the site conditions can effectively reduce the damage to the engineering structure from seismic effects, extend its safe service life, and contribute to the sustainability of the engineering structure. Site conditions are mainly considered in the following aspects: engineering geological conditions and hydrogeological conditions near the surface [3,4], local topographic effects [5,6], and fault site effects [7,8]. The engineering geological conditions near the surface can be investigated from three angles, such as geotechnical type, overburden thickness, and soil structure [9]. Generally speaking, the ground motion on the bedrock site is smaller, followed by the hard soil site, and the soft soil site is the largest under the same ground motion. Meanwhile, the existing analysis shows that the soil structure also has a significant influence on ground motion [10,11]. Specifically, it has been observed that as the burial depth of the hard interlayer increases, both the peak acceleration and the response spectrum of the ground surface increase. Moreover, as the thickness of the hard interlayer increases, the peak acceleration of the ground surface first decreases gradually, then increases gradually, while the response spectrum value increases [12–15]. On the other hand, increasing the burial depth or thickness of the soft interlayer leads to a decrease in both the peak acceleration and the peak of the response spectrum [16,17].

In recent years, people have paid attention to the influence of soft soil on the site's seismic response, and some scholars have carried out in-depth analyses from different perspectives. Xu et al. [18] conducted a study on the seismic damage mechanism of soft soil sites in Fuzhou. The findings indicated that soft soil increases the site excellence period to some extent, and the excellence period is closely related to the stiffness of the foundation soil. Yao et al. [19] suggested that the existence of a local soft interlayer can

significantly amplify or attenuate ground motion, which might affect the lagged spatial consistency between spatially varying ground motions. Cao [20] analyzed the effect of the burial depth of the soft interlayer on the ground motion. The results revealed that the amplification and predominant frequency of the site decreased with the deepening of the soft interlayer location. In a study conducted by Wang et al. [21], the influence of the buried depth of a soft interlayer on ground motion parameters was investigated through site seismic response analysis using the equivalent linear method. The findings revealed that as the burial depth of the soft interlayer increased from shallow to deep, the peak acceleration, peak velocity, and peak response spectrum exhibited a decreasing trend. Additionally, the period corresponding to the maximum value of the characteristic period and response spectrum showed an increasing trend with an increase in the burial depth of the soft interlayer. Tian [22] proposed that under the same Class III site conditions, the presence of soft soil layers makes the ground motion parameter values vary greatly compared to those obtained for sites without soft soil layers. Furthermore, the different locations of the soft soil layers in the soil structure lead to large differences in the ground motion parameter values. Yan et al. [23] focus on the influence of soft interlayer and slope effects on the dynamic response of slope sites through acceleration amplification effects and seismic wave fluctuation mechanisms. They combine the traditional Fourier spectrum and Hilbert marginal spectrum methods to demonstrate the spectral variation characteristics of sites from the frequency domain perspective. Li and Xia et al. [24] calculated three profiles with thicknesses of 3 m, 5 m, and 9 m to analyze the effects of burial depth and thickness of soft soil interlayer on surface ground motion parameters under the condition of constant burial depth. Wang et al. [25] conducted research using an ideal site as the base model and varied the position of the soft soil layer to establish corresponding calculation models. Through soil response analysis, they investigated the effects of the soft interlayer at different locations on parameters such as peak surface acceleration, amplification coefficient, and equivalent shear wave velocity at the site. The study concluded that the influence of the soft interlayer on the site's peak acceleration exhibits an initial amplification followed by a reduction. Additionally, the propagation capacity of the four site types exhibits a certain range, and the equivalent shear wave velocity does not accurately reflect the soil layer structure.

In summary, the soft soil layer has a significant effect on the site's seismic response, especially in the form of a significant increase in the characteristic period. Additionally, compared with the Code [26], the characteristic period of the acceleration response spectrum of a site with a soft soil layer after the regulation is much larger than the value specified in the Code. The method of determining the characteristic period is the key technology in earthquake engineering, and there is little research on the adjustment method of the characteristic period of the site response spectrum. Although a few scholars have conducted relevant research [27–29], the currently available adjustment methods are not intended for sites with soft interlayers. Therefore, a new characteristic period calibration method applicable to soft soil sites has yet to be proposed (i.e., a new seismic engineering technology). In light of these findings, the present study aims to develop soft site models incorporating silt layers, building upon previous research. The influence of the soft soil layer on the seismic response of the site will be analyzed, and a method for adjusting the characteristic period of the response spectrum will be proposed. This research endeavor intends to provide a theoretical foundation for determining the characteristic period of the seismic response spectrum for sites with soft soil layers.

## 2. Ground Motion Input

The input ground motion for the seismic response analysis of a soft site with a silt layer is obtained by artificial synthesis [30–32]. The synthesized ground motions consist of peak accelerations of 50 gal, 100 gal, and 200 gal (1 gal = 1 cm/s$^2$), time intervals of 0.02 s, and discrete points of 2048. The acceleration time range is reduced by half in magnitude as the input ground motions are calculated for the one-dimensional soil seismic response

analytical model. The input ground motion acceleration time range and acceleration response spectrum are drawn in Figure 1, and the characteristic periods of the response spectrum are 0.30 s, 0.35 s, and 0.40 s, respectively.

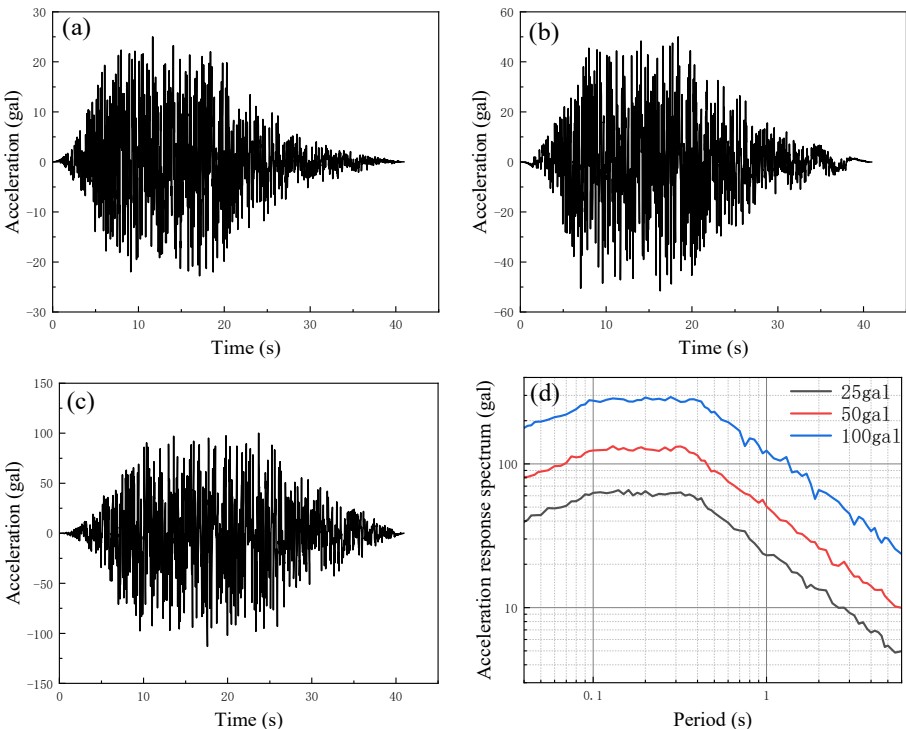

**Figure 1.** Input ground motion: (**a**) acceleration time range (PGA = 25 gal); (**b**) acceleration response spectrum (PGA = 50 gal); (**c**) acceleration time range (PGA = 100 gal); (**d**) acceleration response spectrum.

## 3. Site Seismic Response Analysis

### 3.1. Site Seismic Response Calculation Model and Determination of Dynamic Parameters

For analyzing the seismic response of the site with a soft soil layer, six analytical models are established, each representing a different burial depth of a single layer of silt. Additionally, six analytical models are created, representing different burial depths of two layers of silt. These models are developed based on the survey data and the specific engineering seismic conditions of the site. The objective is to analyze the influence of the characteristics of the thickness and burial depth of the soft soil layer on the site's seismic response.

To analyze the ground motion response of soil layers at the site, it is desirable to have detailed information about the soil profile, including the layered thickness and properties of each soil layer. Additionally, the mechanical properties of the soil also play a crucial role in analyzing the ground motion response. These properties include the shear wave velocity, density, and dynamic nonlinear characteristic parameter values of the soil. Based on the actual field investigation and experimental data and considering different silt layer thicknesses and burial depths, six analytical models with one layer and two layers of silt, each with different burial depths, have been established [33–35]. The profile and mechanical parameters of model 1 are summarized in Table 1. Models 2 to 6 are obtained on the basis of model 1 by varying the number of silty clay layers overlying the silt layer and gradually moving the silty clay from under the silt layer of model 1 to above the silty layer. For instance, model 2 consists of one layer of silty clay with soil class number 3. Model 3 includes two layers of silty clay with class numbers 3 and 4. Model 4 is composed of three layers of silty clay with class numbers 3, 4, and 5. Model 5 is constructed with four layers of silty clay with class numbers 3, 4, 5, and 6. Model 6 comprises five layers of silty clay



with class numbers 3, 4, 5, 6, and 7. The profile and mechanical properties of model 7 are summarized in Table 2. Models 8 to 12 are also obtained by changing the number of layers of overlying silty clay based on model 7. This means that model 8 is overlaid with one layer of silty clay, classified as 3. Model 9 is overlaid with two layers of silty clay, classified as 3 and 4. Model 10 is overlaid with three layers of silty clay, classified as 3, 4, and 5. Model 11 is overlaid with four layers of silty clay, classified as 3, 4, 5, and 6. Lastly, model 12 is overlaid with five layers of silty clay, classified as 3, 4, 5, and 7.

**Table 1.** Analysis model 1.

| No. | Rock-Soil | Soil Class | Depth at the Bottom of Layer (m) | Layer Thickness (m) | Shear Wave Velocity (m/s) | Density (t/m$^3$) |
|---|---|---|---|---|---|---|
| 1 | Silt | 1 | 5.0 | 5.0 | 112 | 1.58 |
| 2 | silty clay | 3 | 9.5 | 4.5 | 160 | 1.86 |
| 3 | silty clay | 4 | 13.0 | 3.5 | 165 | 1.87 |
| 4 | silty clay | 5 | 17.0 | 4.0 | 199 | 1.88 |
| 5 | silty clay | 6 | 21.0 | 4.0 | 212 | 1.96 |
| 6 | silty clay | 7 | 24.0 | 3.0 | 242 | 1.98 |
| 7 | rounded gravel | 8 | 27.0 | 3.0 | 258 | 2.20 |
| 8 | fully weathered andesite | 8 | 30.0 | 3.0 | 393 | 2.25 |
| 9 | bedrock of model | 9 | | | 516 | 2.65 |

**Table 2.** Analysis model 7.

| No. | Rock-Soil | Soil Class | Depth at the Bottom of Layer (m) | Layer Thickness (m) | Shear Wave Velocity (m/s) | Density (t/m$^3$) |
|---|---|---|---|---|---|---|
| 1 | Silt | 1 | 5.0 | 5.0 | 112 | 1.58 |
| 2 | Silt | 2 | 10.0 | 5.0 | 112 | 1.66 |
| 3 | silty clay | 3 | 14.5 | 4.5 | 160 | 1.86 |
| 4 | silty clay | 4 | 18.0 | 3.5 | 165 | 1.87 |
| 5 | silty clay | 5 | 22.0 | 4.0 | 199 | 1.88 |
| 6 | silty clay | 6 | 26.0 | 4.0 | 212 | 1.96 |
| 7 | silty clay | 7 | 29.0 | 3.0 | 242 | 1.98 |
| 8 | rounded gravel | 8 | 32.0 | 3.0 | 258 | 2.20 |
| 9 | fully weathered andesite | 8 | 35.0 | 3.0 | 393 | 2.25 |
| 10 | bedrock of model | 9 | | | 516 | 2.65 |

**Table 3.** Nonlinear parameters of dynamic shear of various soils at different shear strain levels.

| Soil Class | Soil Layer | Modulus Ratio | Shear Strain (10$^{-4}$) | | | | | | | |
|---|---|---|---|---|---|---|---|---|---|---|
| | | Damping Ratio | 0.05 | 0.1 | 0.5 | 1 | 5 | 10 | 50 | 100 |
| 1 | silt | $G/G_{max}$ | 0.9902 | 0.98086 | 0.9105 | 0.8358 | 0.5045 | 0.3374 | 0.0923 | 0.0483 |
| | | $\zeta$ | 0.0173 | 0.0244 | 0.0525 | 0.0711 | 0.1236 | 0.1429 | 0.1672 | 0.1712 |
| 2 | silt | $G/G_{max}$ | 0.9913 | 0.9827 | 0.9189 | 0.8500 | 0.5313 | 0.3617 | 0.1018 | 0.0536 |
| | | $\zeta$ | 0.0088 | 0.0135 | 0.0356 | 0.0525 | 0.1073 | 0.1303 | 0.1615 | 0.1669 |
| 3 | silty clay | $G/G_{max}$ | 0.9918 | 0.9838 | 0.9241 | 0.8588 | 0.5489 | 0.3783 | 0.1085 | 0.0573 |
| | | $\zeta$ | 0.0138 | 0.0199 | 0.0459 | 0.0641 | 0.1201 | 0.1428 | 0.1735 | 0.1788 |
| 4 | silty clay | $G/G_{max}$ | 0.9925 | 0.9851 | 0.9296 | 0.8684 | 0.5689 | 0.3975 | 0.1166 | 0.0619 |
| | | $\zeta$ | 0.0123 | 0.0176 | 0.0402 | 0.0561 | 0.1053 | 0.1258 | 0.1542 | 0.1592 |
| 5 | silty clay | $G/G_{max}$ | 0.9939 | 0.9878 | 0.9419 | 0.8903 | 0.6187 | 0.4479 | 0.1396 | 0.0750 |
| | | $\zeta$ | 0.0157 | 0.0218 | 0.0461 | 0.0626 | 0.1136 | 0.1356 | 0.1677 | 0.1736 |
| 6 | silty clay | $G/G_{max}$ | 0.9943 | 0.9887 | 0.9460 | 0.8975 | 0.6365 | 0.4668 | 0.1490 | 0.0805 |
| | | $\zeta$ | 0.0181 | 0.0249 | 0.0512 | 0.0688 | 0.1234 | 0.1473 | 0.1827 | 0.1894 |
| 7 | silty clay | $G/G_{max}$ | 0.9950 | 0.9901 | 0.9524 | 0.9092 | 0.6669 | 0.5003 | 0.1668 | 0.0910 |
| | | $\zeta$ | 0.0106 | 0.0152 | 0.0342 | 0.0478 | 0.0936 | 0.1154 | 0.1504 | 0.157 |
| 8 | Rounded gravel and pebbles | $G/G_{max}$ | 0.990 | 0.970 | 0.900 | 0.850 | 0.700 | 0.550 | 0.320 | 0.200 |
| | | $\zeta$ | 0.004 | 0.006 | 0.019 | 0.030 | 0.075 | 0.090 | 0.110 | 0.120 |
| 9 | bedrock | $G/G_{max}$ | 1.000 | 1.000 | 1.000 | 1.000 | 1.000 | 1.000 | 1.000 | 1.000 |
| | | $\zeta$ | 0.004 | 0.008 | 0.01 | 0.015 | 0.021 | 0.030 | 0.036 | 0.046 |

The dynamic nonlinear parameters and density values of each soil layer in different calculation site models are derived from the experimental results of the seismic safety evaluation project at the actual engineering site. The dynamic nonlinear parameters and

density values of each soil layer are listed in Table 3, and the density values are listed in Tables 1 and 2.

### 3.2. Analysis of Calculation Results

When the equivalent linearization method [36,37] is adopted for the site seismic response calculation of each analytical model, the input ground motion acceleration time histories corresponding to three peak acceleration levels (25 gal, 50 gal, and 100 gal) are taken as the computational base incident ground motion of the one-dimensional soil response analysis model. The ground motion acceleration time histories and response spectrum values of the surface horizontal seismic response are obtained through the computational analysis of the horizontal seismic response of each analytical model.

The peak acceleration of surface horizontal seismic response under different ground motions for each analytical model is shown in Table 4, from which the dynamic amplification coefficients of surface horizontal seismic response for each analytical model are obtained and presented in Table 5.

**Table 4.** The peak acceleration of each analytical model.

| Input Peak Acceleration/gal | | | | Input Peak Acceleration/gal | | | |
|---|---|---|---|---|---|---|---|
| **Surface Peak Acceleration/gal** | **25** | **50** | **100** | **Surface Peak Acceleration/gal** | **25** | **50** | **100** |
| **Analytical Model** | | | | **Analytical Model** | | | |
| 1 | 53.1 | 97.1 | 193.2 | 7 | 48.9 | 96 | 174.2 |
| 2 | 43.8 | 86.5 | 152.3 | 8 | 38.7 | 62.6 | 110.1 |
| 3 | 41.2 | 72.4 | 118.7 | 9 | 33.7 | 55.2 | 102.8 |
| 4 | 38.7 | 58.2 | 109.9 | 10 | 31.1 | 53.9 | 83.9 |
| 5 | 36.4 | 52.6 | 100.6 | 11 | 30.4 | 48.2 | 69.4 |
| 6 | 30.6 | 45 | 87.4 | 12 | 29.6 | 46.4 | 64.6 |

**Table 5.** Dynamic amplification coefficient of the surface seismic response of each analytical model.

| Input Peak Acceleration/gal | | | | Input Peak Acceleration/gal | | | |
|---|---|---|---|---|---|---|---|
| **Dynamic Amplification Coefficient** | **25** | **50** | **100** | **Dynamic Amplification Coefficient** | **25** | **50** | **100** |
| **Analytical Model** | | | | **Analytical Model** | | | |
| 1 | 2.124 | 1.942 | 1.932 | 7 | 1.956 | 1.92 | 1.742 |
| 2 | 1.752 | 1.73 | 1.523 | 8 | 1.548 | 1.252 | 1.101 |
| 3 | 1.648 | 1.448 | 1.187 | 9 | 1.348 | 1.104 | 1.028 |
| 4 | 1.548 | 1.164 | 1.099 | 10 | 1.244 | 1.078 | 0.839 |
| 5 | 1.456 | 1.052 | 1.006 | 11 | 1.216 | 0.964 | 0.694 |
| 6 | 1.224 | 0.900 | 0.874 | 12 | 1.184 | 0.928 | 0.646 |

From Table 4, it can be seen that under the same input peak acceleration level, the thicker the soft soil layer, the smaller the surface peak acceleration; the deeper the soft soil layer is buried, the smaller the surface peak acceleration.

As can be seen from Table 5, at the same input peak acceleration level, the thicker the soft soil layer, the smaller the dynamic amplification coefficient of surface peak acceleration; the deeper the soft soil layer is buried, the smaller the dynamic amplification coefficient of surface peak acceleration; and the attenuation of the dynamic amplification coefficient is slower as the burial depth increases. Under the same thickness and burial depth of the soft soil layer, with the increase in input peak acceleration, the dynamic amplification coefficient of surface peak acceleration gradually decreases, which indicates that the site soil has significant nonlinearity.

The variation of peak ground acceleration with different burial depths of the soft soil layer is given according to Table 4, as depicted in Figure 2. Likewise, the variation of the peak ground acceleration dynamic amplification coefficient with different burial depths of the soft soil layer is given according to Table 5, illustrated in Figure 3.

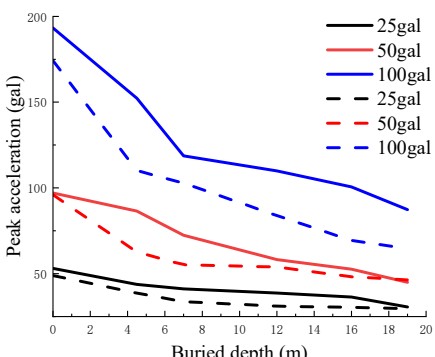

**Figure 2.** Variation characteristics of peak ground acceleration with different buried depths of soft soil layers (solid lines for models with one silt layer, dashed lines for models with two silt layers).

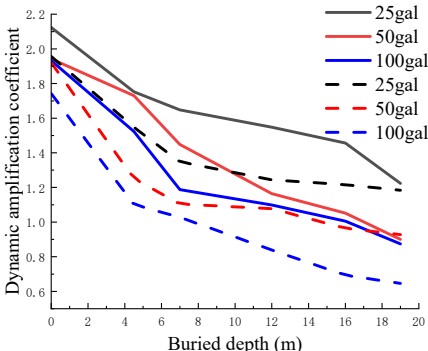

**Figure 3.** Variation characteristics of peak acceleration dynamic amplification coefficient with different buried depths of soft soil layer (solid lines for models with one silt layer, dashed lines for models with two silt layers).

From Figure 2, it can be seen that for a given input ground motion level, the peak ground acceleration decreases as the burial depth of the soft soil layer increases. The attenuation of the peak ground acceleration is more pronounced near the surface, and it becomes slower as the burial depth increases. Moreover, for different input ground motion levels, higher input peak accelerations result in faster attenuation of the peak ground acceleration. Similarly, the thickness of the soft soil layer also affects peak ground acceleration. A thicker layer of soft soil results in a smaller peak ground acceleration. The attenuation of the peak ground acceleration is faster near the shallow surface in thicker soil layers. This difference becomes more obvious as the input ground motion peak acceleration increases.

As seen in Figure 3, under the same input ground motion, the dynamic amplification coefficient of the peak surface acceleration is smaller when the burial depth of the soft soil layer increases. The attenuation of the dynamic amplification coefficient is faster near the surface and slower as the burial depth increases. Furthermore, for different input ground motion levels, the dynamic amplification coefficient attenuates faster as the input peak acceleration increases. Meanwhile, the thickness of different soft soil layers affects the peak surface acceleration. A thicker layer leads to a smaller dynamic amplification coefficient and faster attenuation near the shallow surface. The difference in the dynamic amplification coefficient becomes more obvious with an increase in the input ground motion peak acceleration.

The site-related response spectra for the damping ratio of 5% are also obtained in the seismic response analysis of each model, as shown in Figures 4 and 5.

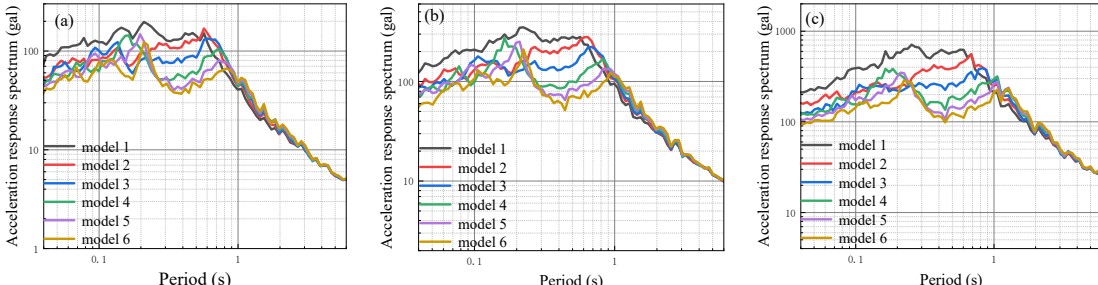

**Figure 4.** Site-related acceleration response spectra of each analysis model with a layer of silt at different input ground motions: (**a**) Input ground motion peak acceleration of 25 gal; (**b**) input ground motion peak acceleration of 50 gal; (**c**) input ground motion peak acceleration of 100 gal.

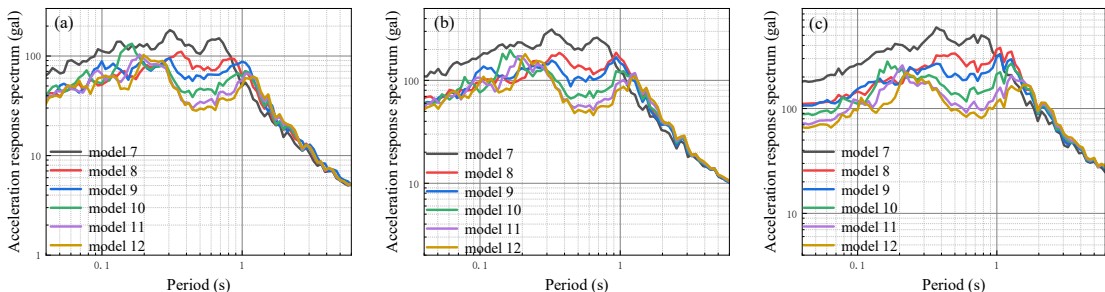

**Figure 5.** Site-related acceleration response spectra of each analysis model with two layers of silt at different input ground motions: (**a**) Input ground motion peak acceleration of 25 gal; (**b**) input ground motion peak acceleration of 50 gal; (**c**) input ground motion peak acceleration of 100 gal.

Figure 4 illustrates that in the model with one silt layer, at the same input ground motion peak acceleration level, the overall trend within the fluctuation band of the site-related response spectrum is that the acceleration response spectrum value decreases as the burial depth of the soft soil layer increases. Additionally, the initial frequency of the response spectrum attenuation section decreases, and the dominant frequency band of the response spectrum becomes wider. The variation trend of the response spectrum for each model is generally consistent at different input peak acceleration levels, while the greater the input peak acceleration, the larger the response spectrum value. The response spectra of models are close to each other for the periodic acceleration spectrum above 1 s. This suggests that the burial depth of the soft soil layer has less influence on the long-period ground motion.

Figure 5 demonstrates that in the model with two silt layers, at the same input ground motion peak acceleration level, the overall performance within the fluctuation band of the site-related response spectrum is that the acceleration response spectrum value decreases as the burial depth of the soft soil layer increases. Additionally, the initial frequency of the attenuation section decreases, and the dominant frequency band of the response spectrum becomes wider. The trend of variation in the response spectrum for each model remains generally consistent at different input peak acceleration levels. However, it is observed that as the input peak acceleration increases, the response spectrum values also increase. The response spectra of models are close to each other for the periodic acceleration spectrum above 1s, which indicates that the burial depth of the soft soil layer has less influence on the long-period ground motion.

Based on the free surface horizontal acceleration response spectrum of the engineering site, the site-related normalized response spectrum with a damping ratio of 5% is calculated and displayed in Figures 6 and 7.

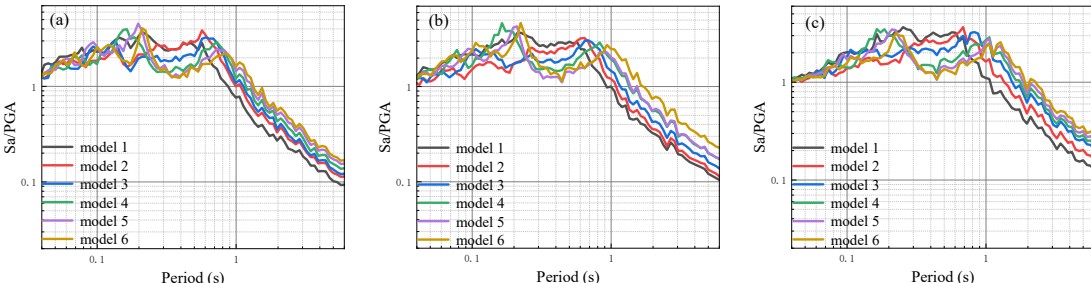

**Figure 6.** Site-related normalized spectrum of each calculation model with a layer of silt under different input ground motions: (**a**) Input ground motion peak acceleration of 25 gal; (**b**) input ground motion peak acceleration of 50 gal; (**c**) input ground motion peak acceleration of 100 gal.

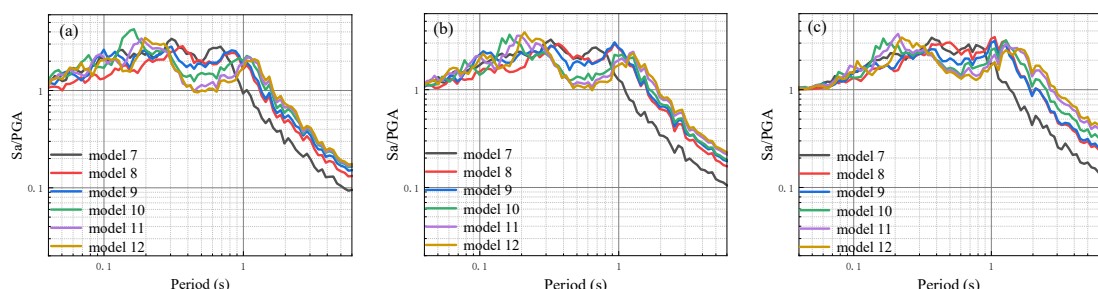

**Figure 7.** Site-related normalized spectrum of each calculation model with two layers of silt under different input ground motions: (**a**) Input ground motion peak acceleration of 25 gal; (**b**) input ground motion peak acceleration of 50 gal; (**c**) input ground motion peak acceleration of 100 gal.

As can be seen from Figure 6, for the calculation model containing one silt layer, the normalized spectrum of site correlation varies with the burial depth of the soft layer at the same input ground motion level. Within the normalized spectrum fluctuation band, the general performance is that the normalized spectrum value is lower the deeper the burial depth of the soft soil layer. Additionally, the initial frequency of the attenuation section of the normalized spectrum decreases, and the dominant band of the normalized spectrum becomes wider. The variation trend of the normalized spectrum for each model remains approximately the same under different input ground motion levels. However, it is noted that as the input peak acceleration increases, the normalized spectrum tends to decrease. In the normalized spectrum of 0.1 s and below, the results of the models are close to each other, indicating that the burial depth of the soft soil layer has less influence on the normalized spectrum of a short period. Meanwhile, in the period above 0.1 s, the burial depth of the soft soil layer has more influence on the normalized spectrum. The general performance is that the deeper the burial depth of the soft soil layer, the larger the normalized spectrum value and the difference between them becomes more noticeable.

It can also be seen from Figure 7 that for the calculation model containing two silt layers, the normalized spectrum of site correlation varies with the burial depth of the soft layer under the same input ground motion level. In the normalized spectrum fluctuation band, the general performance is that the normalized spectrum value decreases with increasing the burial depth of the soft soil layer. Additionally, the initial frequency of the attenuation section of the normalized spectrum decreases, and the dominant band of the normalized spectrum becomes wider as the burial depth of the soft soil layer increases. The variation trend of the normalized spectrum of the models remains generally consistent under different input ground motion levels. However, it is observed that the normalized spectrum tends to decrease as the input peak acceleration increases. In the normalized spectrum within the range of 0.1 s and below, the results of models show a similar trend, indicating that the burial depth of the soft soil layer has less influence on the normalized spectrum over a short period. However, in the period above 0.1 s, the burial depth of the

soft soil layer has more influence on the normalized spectrum. The overall performance is that the deeper the burial depth of the soft soil layer, the larger the normalized spectrum value and the difference between the models becomes more pronounced.

## 4. A Method for Adjusting the Characteristic Period of Response Spectrum

According to the site category determination method [38] of the Code [26] in China, the site categories of each analytical model can be obtained, as summarized in Table 6.

**Table 6.** Site categories of analytical models.

| Analytical Model | Overburden Thickness (m) | Equivalent Shear Wave Velocity (m/s) | Site Category | Analytical Model | Overburden Thickness (m) | Equivalent Shear Wave Velocity (m/s) | Site Category |
|---|---|---|---|---|---|---|---|
| 1 | 30 | 156.0 | II | 7 | 35 | 134.5 | III |
| 2 | 30 | 156.0 | II | 8 | 35 | 134.5 | III |
| 3 | 30 | 156.0 | II | 9 | 35 | 134.5 | III |
| 4 | 30 | 156.0 | II | 10 | 35 | 142.0 | III |
| 5 | 30 | 161.3 | II | 11 | 35 | 161.3 | II |
| 6 | 30 | 182.4 | II | 12 | 35 | 182.4 | II |

From Table 6, it can be seen that the analyzed model site categories in this paper are II and III. Combined with Table 5.1.4-2 of the Code [26], the characteristic period of the site is 0.45 s (corresponding to Class II sites) or 0.65 s (corresponding to Class III sites) if considered following the third design seismic grouping.

The regularized response spectrum of the sites [39] is obtained following the format of the regularized response spectrum specified in the Code [26], which is presented in Figures 8–13. Additionally, the characteristic period of the response spectrum of each model is given in Table 7.

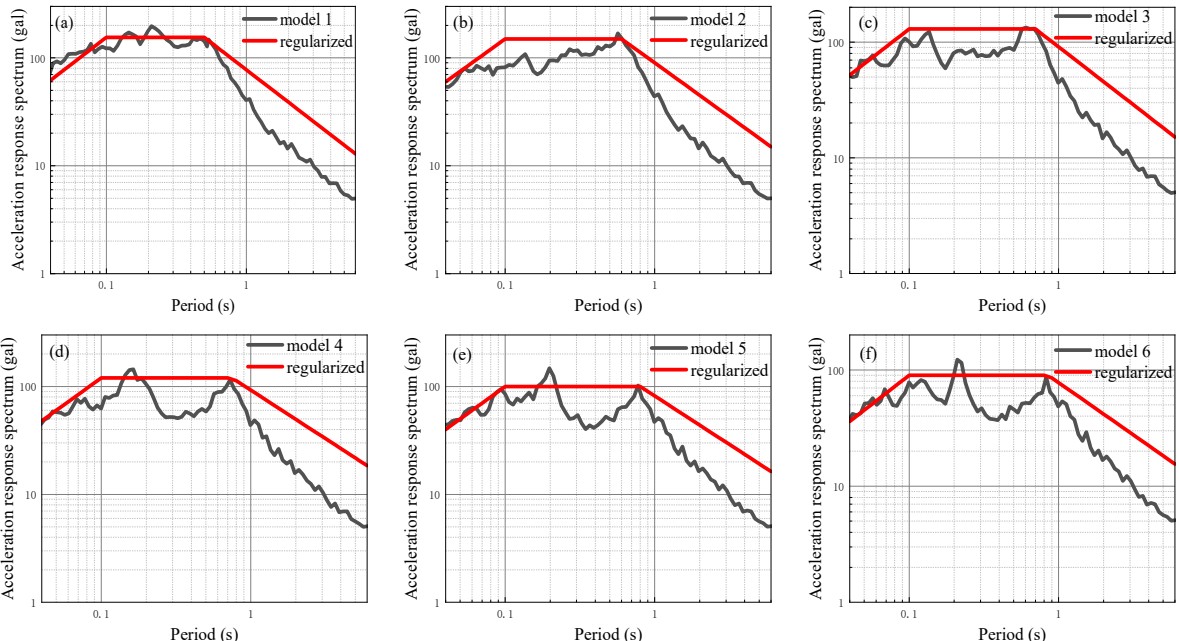

**Figure 8.** Site acceleration response spectrum of each analysis model with a layer of silt under input ground motion with a peak acceleration of 25 gal: (**a**–**f**) corresponding models 1 to 6.

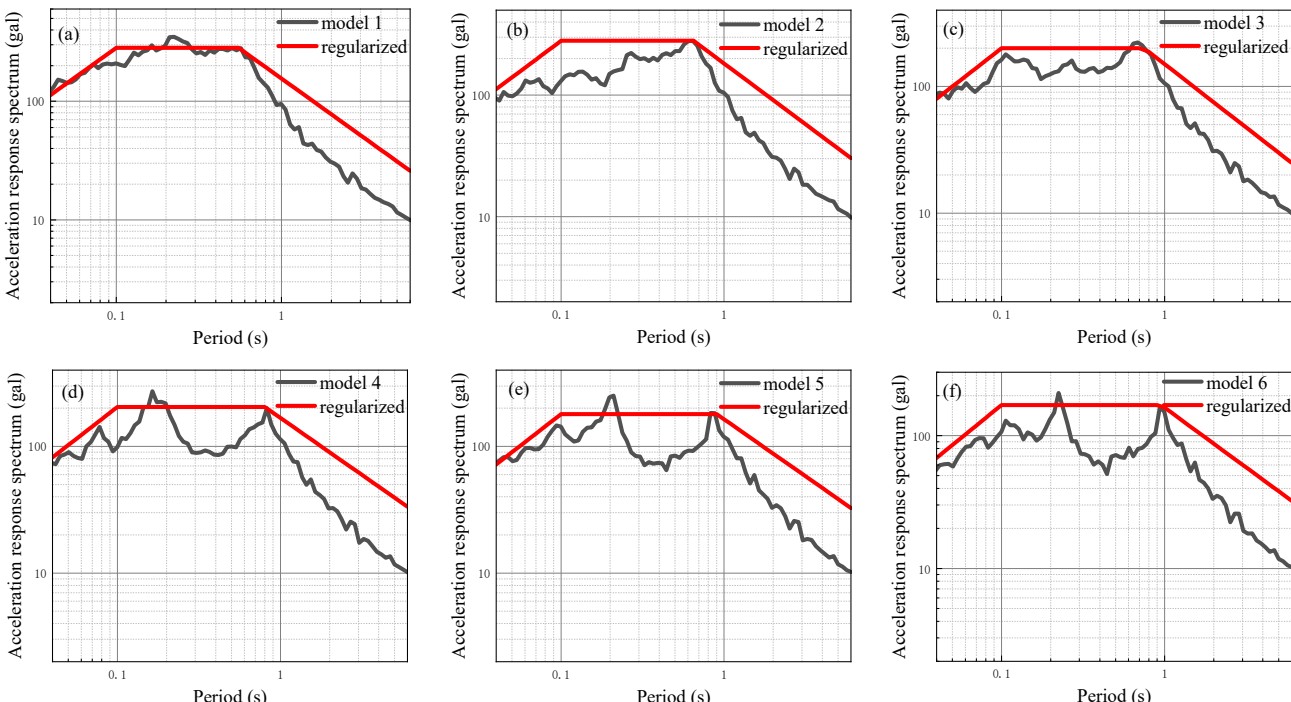

**Figure 9.** Site acceleration response spectrum of each analysis model with a layer of silt under input ground motion with a peak acceleration of 50 gal: (**a**–**f**) corresponding models 1 to 6.

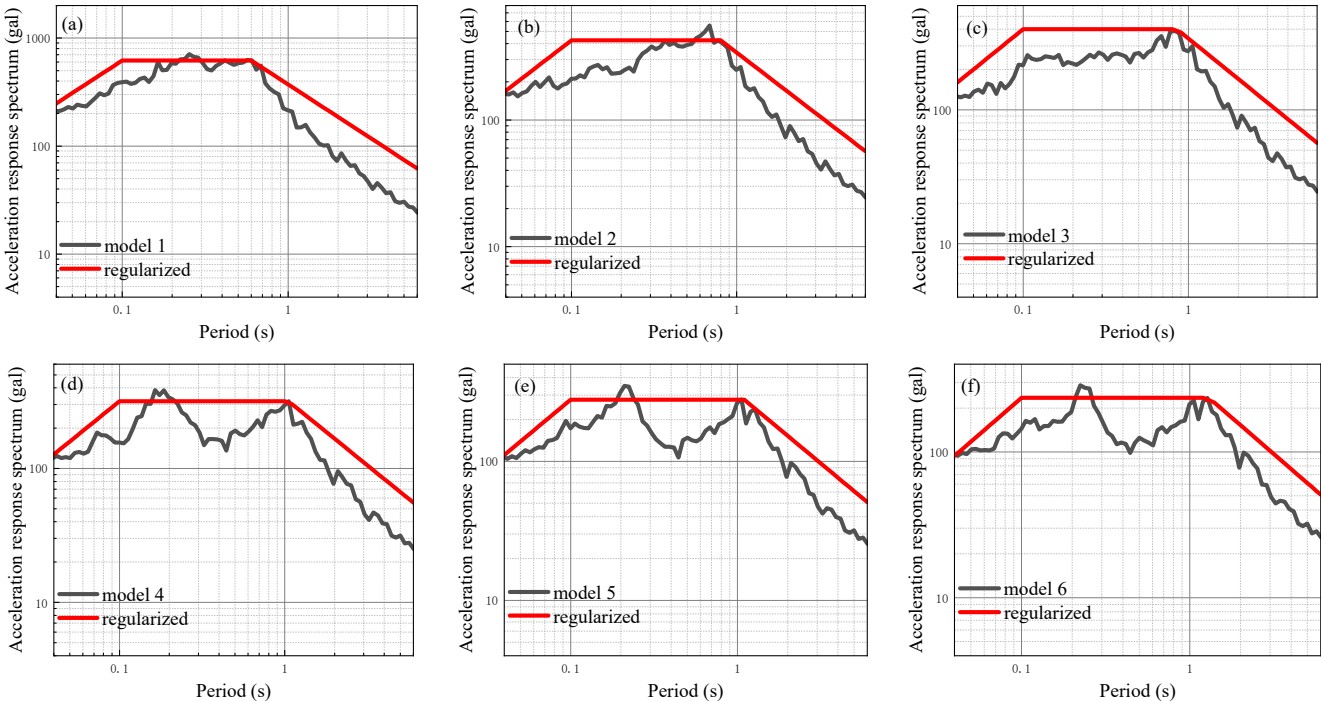

**Figure 10.** Site acceleration response spectrum of each analysis model with a layer of silt under input ground motion with a peak acceleration of 100 gal: (**a**–**f**) corresponding models 1 to 6.

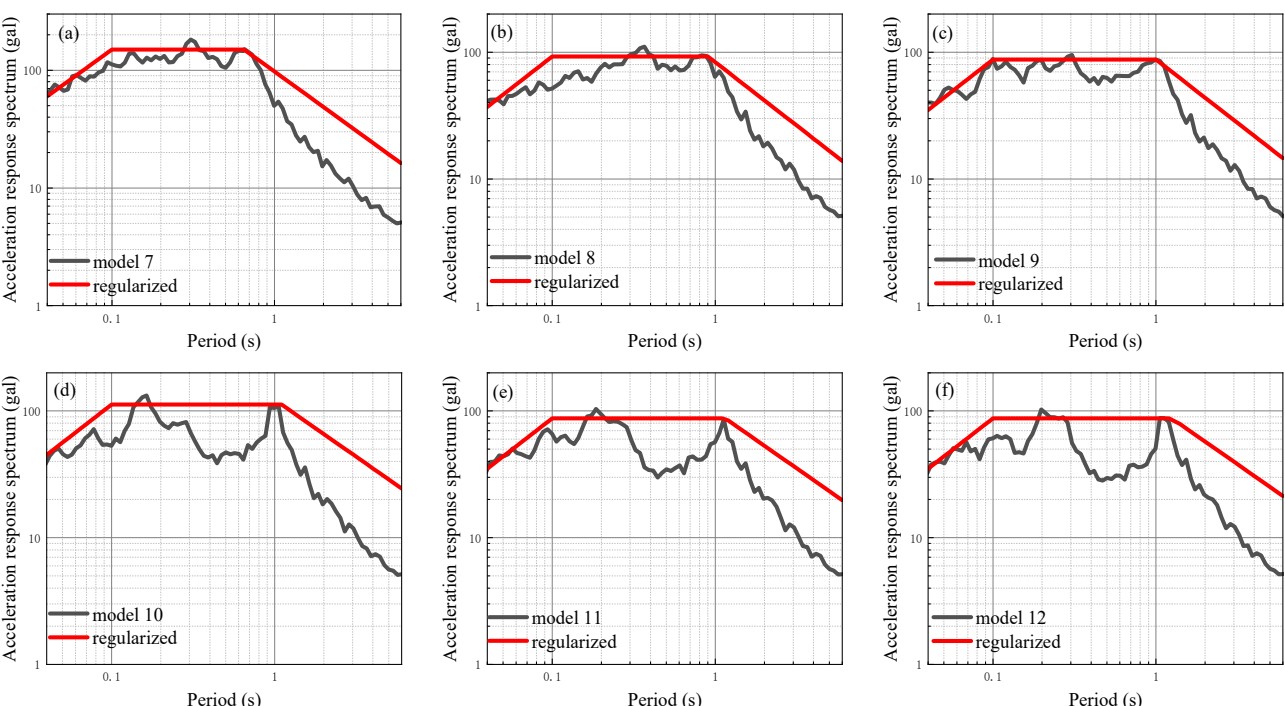

**Figure 11.** Site acceleration response spectrum of each analysis model containing two layers of silt under input ground motion with a peak acceleration of 25 gal: (**a**–**f**) corresponding models 7 to 12.

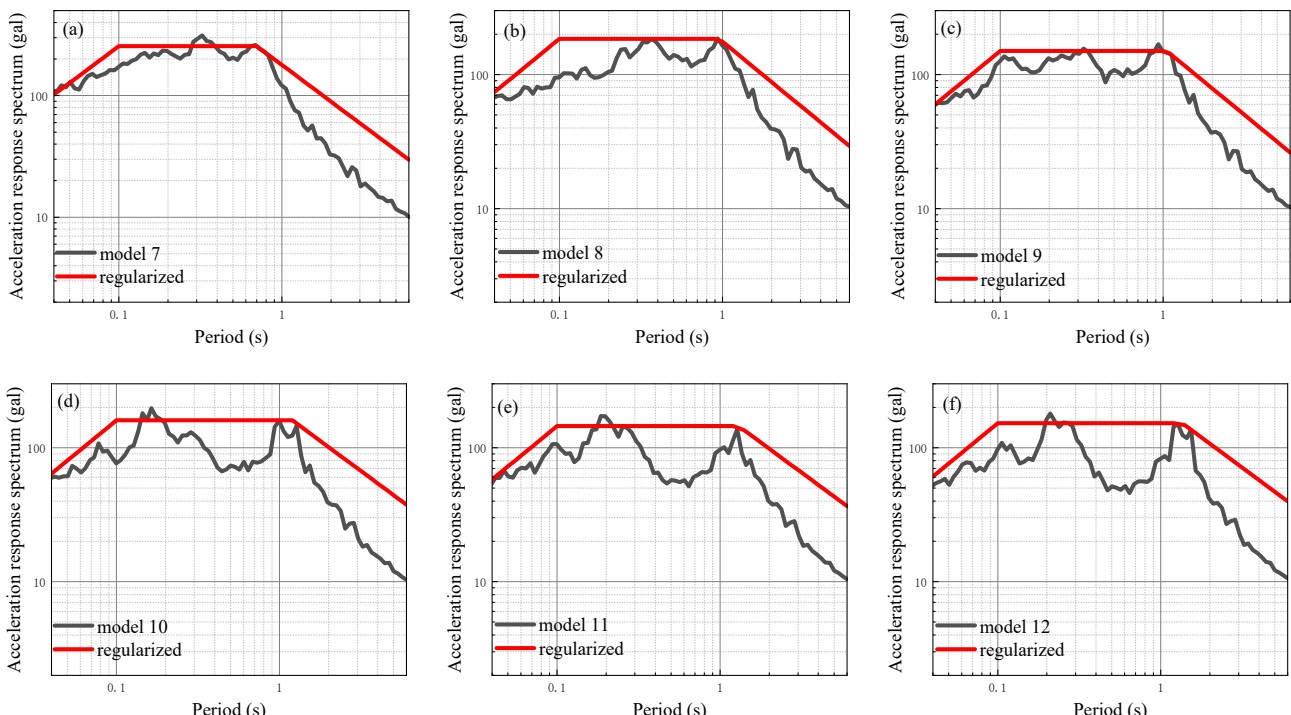

**Figure 12.** Site acceleration response spectrum of each analysis model containing two layers of silt under input ground motion with a peak acceleration of 50 gal: (**a**–**f**) corresponding models 7 to 12.

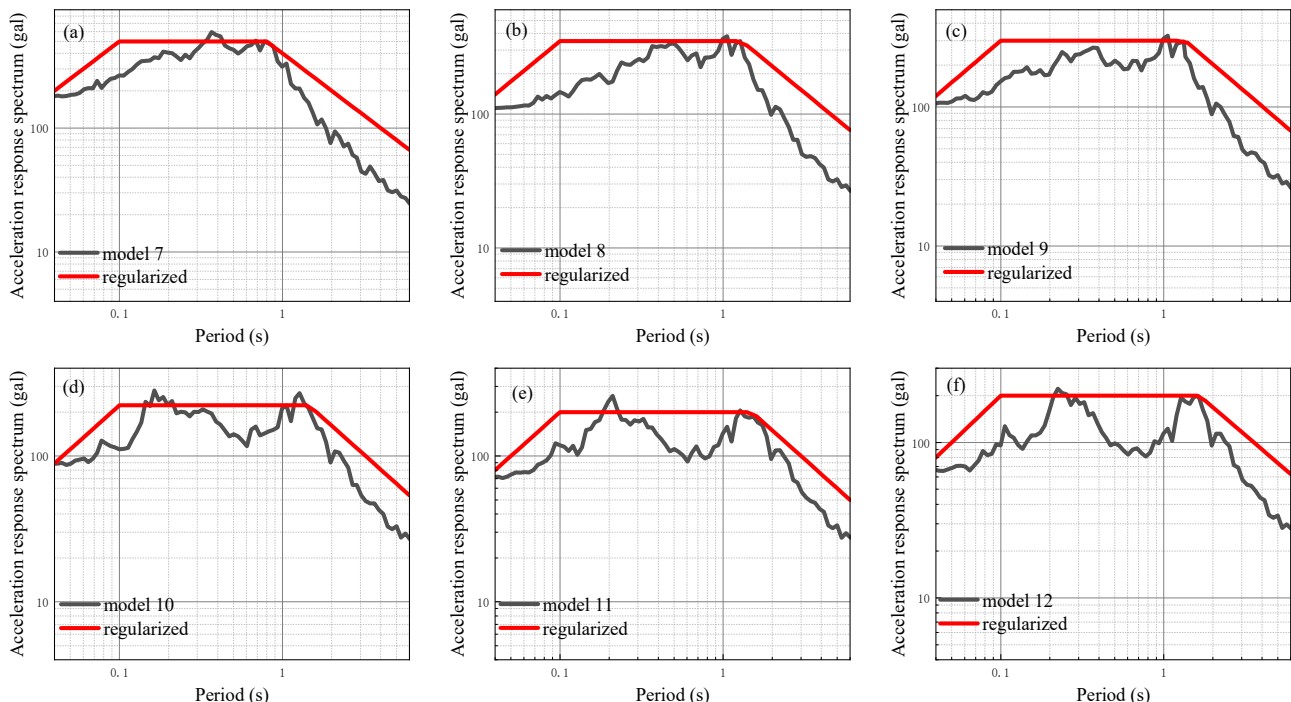

**Figure 13.** Site acceleration response spectrum of each analysis model containing two layers of silt under input ground motion with a peak acceleration of 100 gal: (**a**–**f**) corresponding models 7 to 12.

**Table 7.** The characteristic periodicity of the regularized site response spectrum of each analytical model.

| Input Peak Acceleration/gal | | | | Input Peak Acceleration/gal | | | |
|---|---|---|---|---|---|---|---|
| Characteristic Period/s | 25 | 50 | 100 | Characteristic Period/s | 25 | 50 | 100 |
| Analytical Model | | | | Analytical Model | | | |
| 1 | 0.5 | 0.55 | 0.6 | 7 | 0.7 | 0.75 | 0.95 |
| 2 | 0.6 | 0.65 | 0.8 | 8 | 0.9 | 0.95 | 1.2 |
| 3 | 0.7 | 0.75 | 0.85 | 9 | 1 | 1.05 | 1.35 |
| 4 | 0.75 | 0.8 | 1.05 | 10 | 1.1 | 1.2 | 1.45 |
| 5 | 0.8 | 0.9 | 1.1 | 11 | 1.15 | 1.3 | 1.5 |
| 6 | 0.85 | 0.95 | 1.2 | 12 | 1.25 | 1.35 | 1.65 |

From Figures 8–13 and Table 7, it can be seen that at the same input peak acceleration level, there is a trend where the response spectrum characteristic period increases with the thicker soft soil layer thickness. Additionally, the characteristic period of the response spectrum also increases with the deeper burial of the soft soil layer. Moreover, it can be observed that as the burial depth increases, the rate of increase in the response spectrum characteristic period gradually decreases, while the rate of increase near the shallow surface is faster. Additionally, the response spectrum characteristic period gradually increases with the increase in input peak acceleration while keeping the soft soil thickness and burial depth constant.

In order to further analyze the impact of different input peak acceleration levels on the seismic response of the site with a soft soil layer, additional seismic responses are calculated using input peak accelerations of 50 gal, 100 gal, 150 gal, 200 gal, and 300 gal. These calculations are performed by modulating the ground motion time history of 25 gal using amplitude modulation. The results of the one-dimensional site seismic response analysis of models 1 to 6 are presented in Figures 14–19, which show the site-related acceleration response spectra. On this basis, the calculated site-related acceleration response spectrum is regularized according to the aforementioned method, leading to the regularized response

spectrum and related parameters. The regularized spectrum can be found in Figures 14–19. Additionally, the characteristic period of the regularized spectrum is summarized in Table 8.

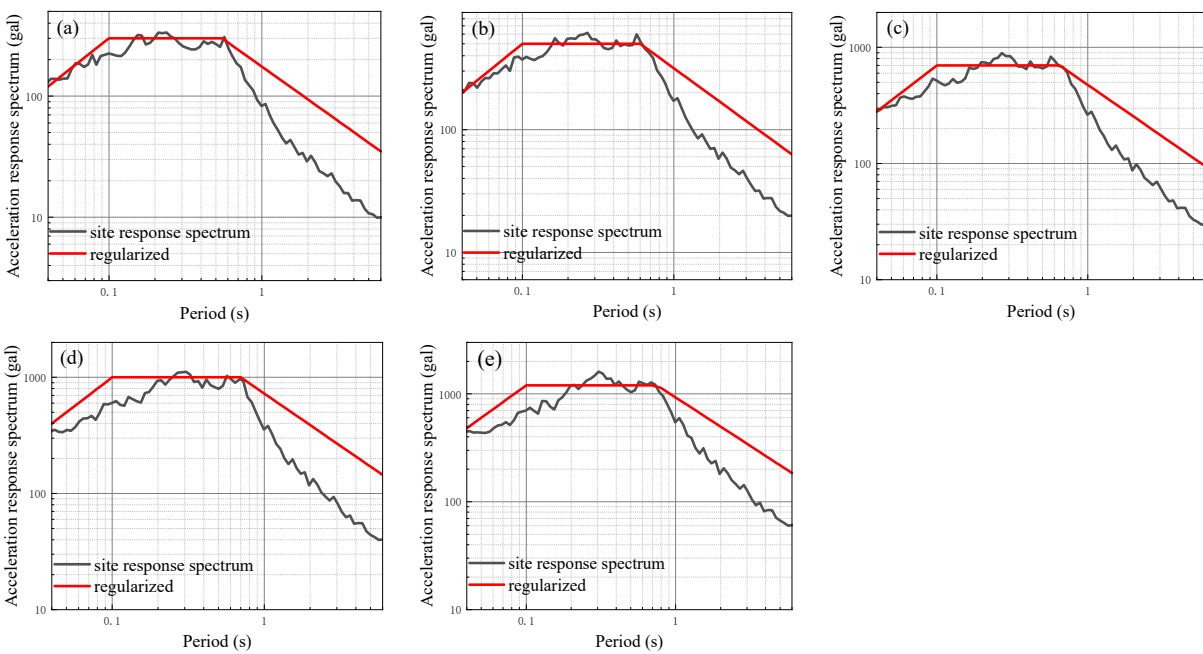

**Figure 14.** Site-related acceleration response spectrum of model 1 under different input levels: (**a**) Input ground motion peak acceleration of 50 gal; (**b**) input ground motion peak acceleration of 100 gal; (**c**) input ground motion peak acceleration of 150 gal; (**d**) input ground motion peak acceleration of 200 gal; (**e**) input ground motion peak acceleration of 300 gal.

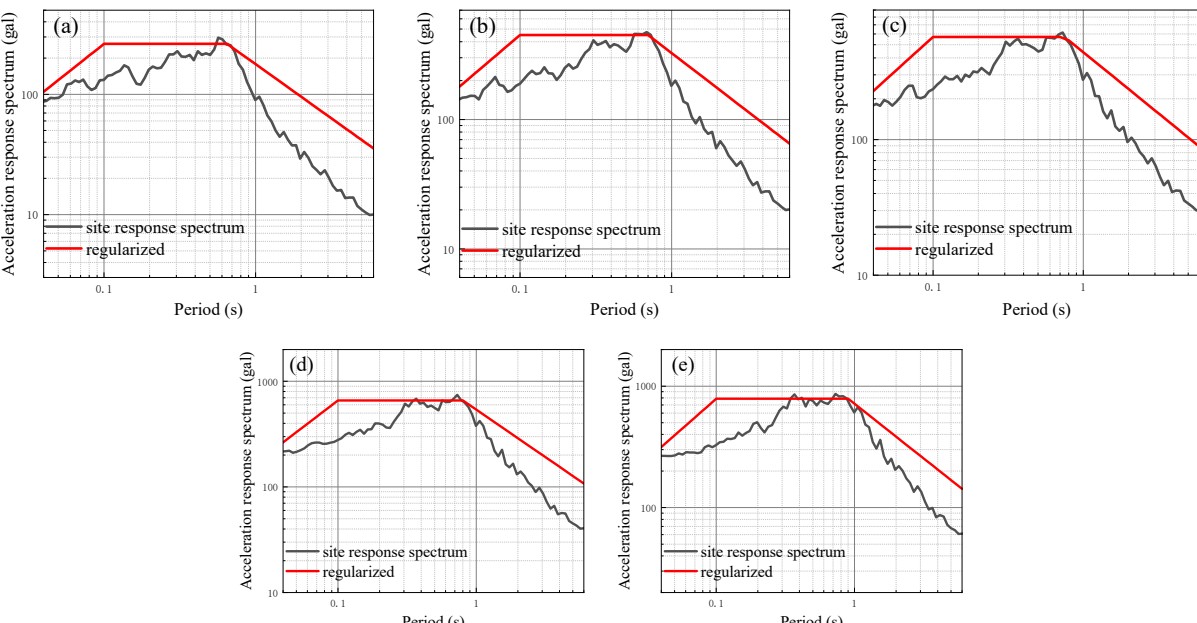

**Figure 15.** Site-related acceleration response spectrum of model 2 under different input levels: (**a**) Input ground motion peak acceleration of 50 gal; (**b**) input ground motion peak acceleration of 100 gal; (**c**) input ground motion peak acceleration of 150 gal; (**d**) input ground motion peak acceleration of 200 gal; (**e**) input ground motion peak acceleration of 300 gal.

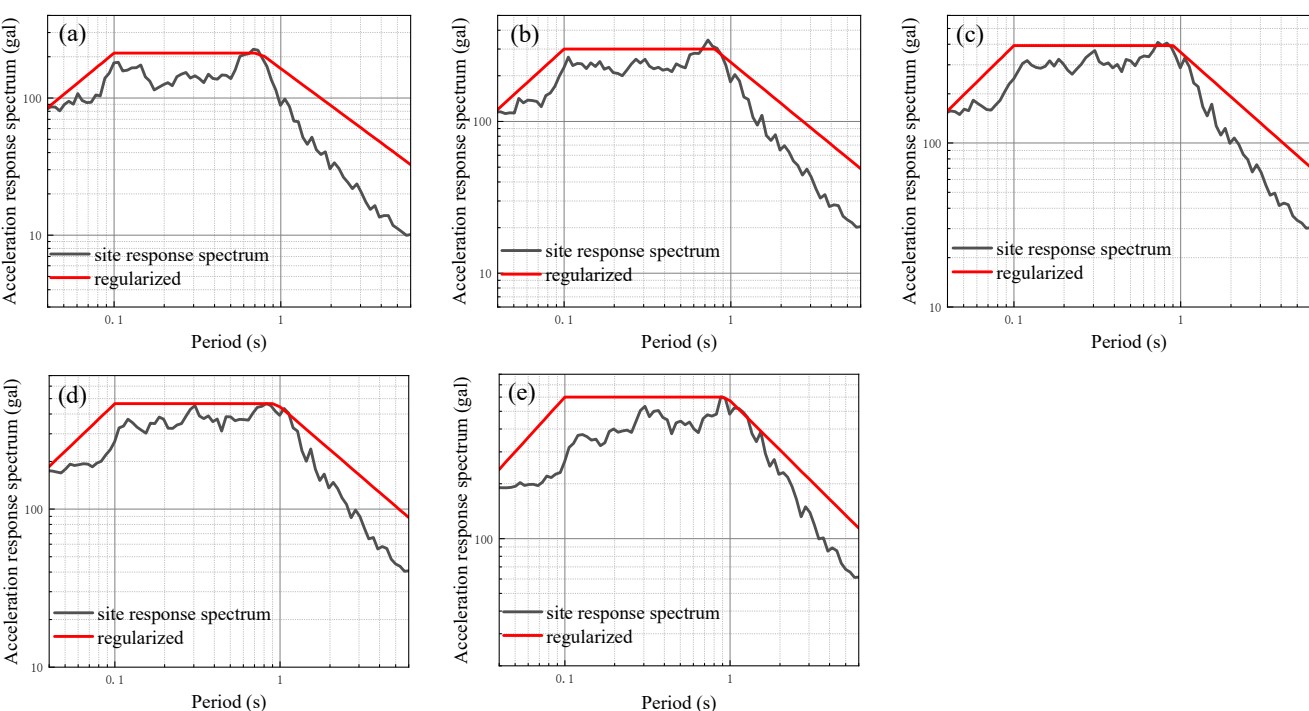

**Figure 16.** Site-related acceleration response spectrum of model 3 under different input levels: (**a**) Input ground motion peak acceleration of 50 gal; (**b**) input ground motion peak acceleration of 100 gal; (**c**) input ground motion peak acceleration of 150 gal; (**d**) input ground motion peak acceleration of 200 gal; (**e**) input ground motion peak acceleration of 300 gal.

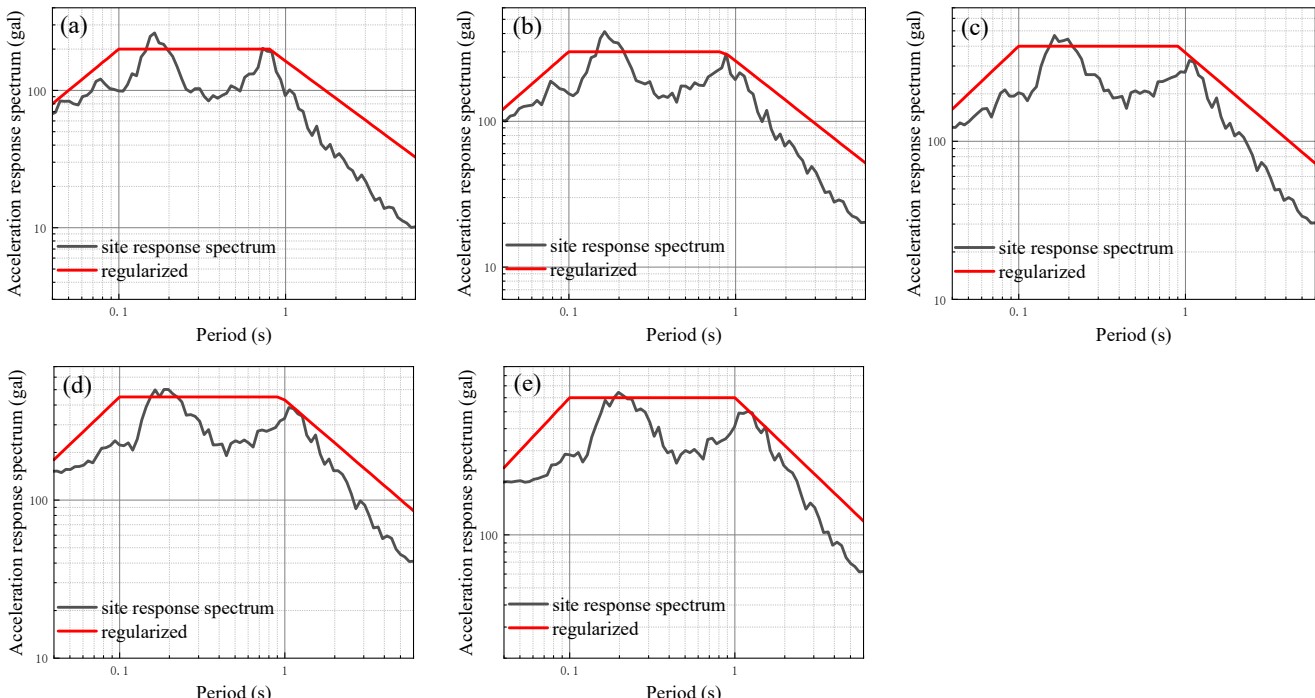

**Figure 17.** Site-related acceleration response spectrum of model 4 under different input levels: (**a**) Input ground motion peak acceleration of 50 gal; (**b**) input ground motion peak acceleration of 100 gal; (**c**) input ground motion peak acceleration of 150 gal; (**d**) input ground motion peak acceleration of 200 gal; (**e**) input ground motion peak acceleration of 300 gal.

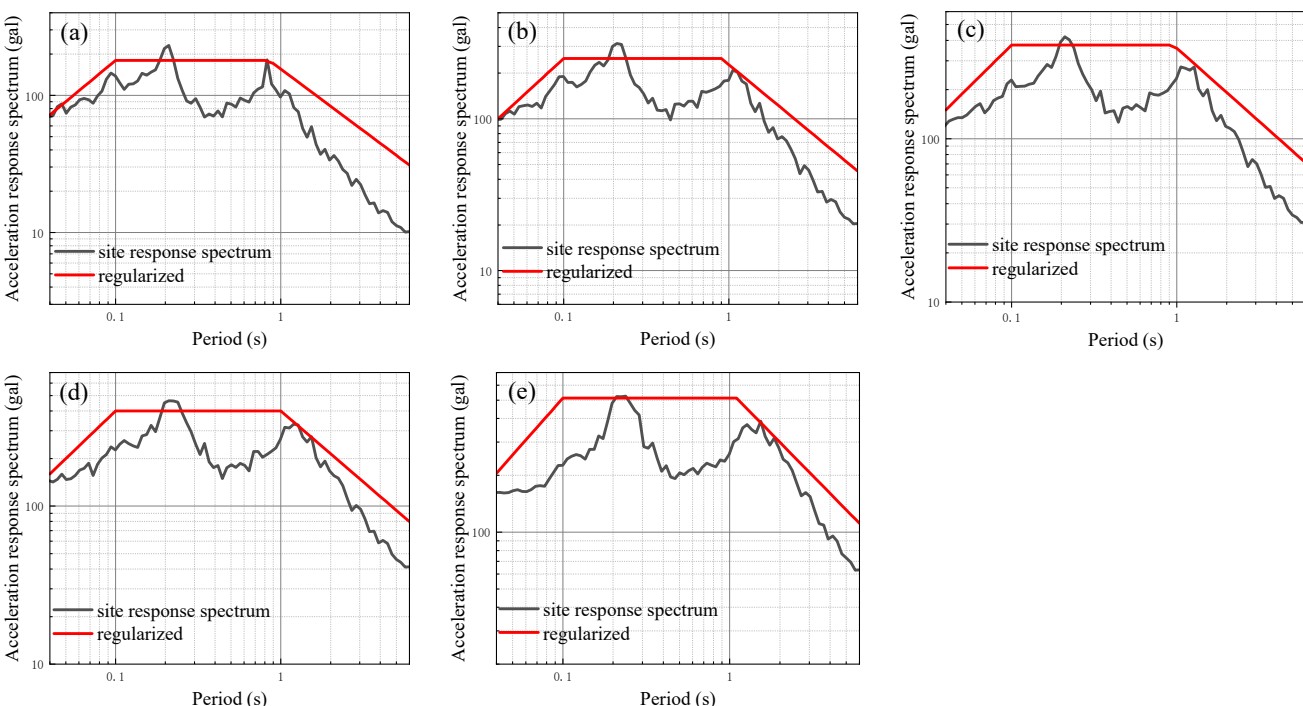

**Figure 18.** Site-related acceleration response spectrum of model 5 under different input levels: (**a**) Input ground motion peak acceleration of 50 gal; (**b**) input ground motion peak acceleration of 100 gal; (**c**) input ground motion peak acceleration of 150 gal; (**d**) input ground motion peak acceleration of 200 gal; (**e**) input ground motion peak acceleration of 300 gal.

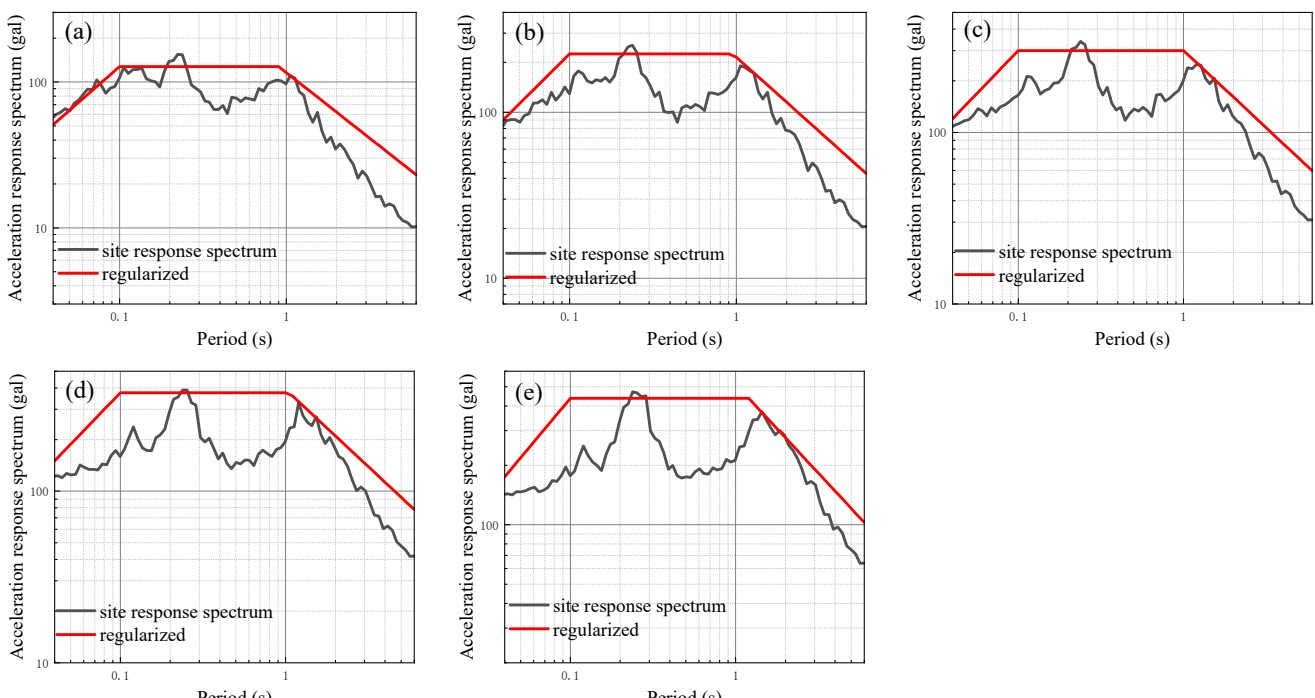

**Figure 19.** Site-related acceleration response spectrum of model 6 under different input levels: (**a**) Input ground motion peak acceleration of 50 gal; (**b**) input ground motion peak acceleration of 100 gal; (**c**) input ground motion peak acceleration of 150 gal; (**d**) input ground motion peak acceleration of 200 gal; (**e**) input ground motion peak acceleration of 300 gal.

**Table 8.** The characteristic periods of models 1–6 at different input levels.

| Analytical Model | | | | | | |
|---|---|---|---|---|---|---|
| Characteristic Period/s | 1 | 2 | 3 | 4 | 5 | 6 |
| Input Peak Acceleration/gal | | | | | | |
| 25 * | 0.50 | 0.6 | 0.7 | 0.75 | 0.8 | 0.85 |
| 50 | 0.55 | 0.65 | 0.75 | 0.8 | 0.85 | 0.9 |
| 100 | 0.60 | 0.7 | 0.8 | 0.85 | 0.9 | 0.95 |
| 150 | 0.65 | 0.75 | 0.85 | 0.9 | 0.95 | 1.0 |
| 200 | 0.70 | 0.8 | 0.9 | 0.95 | 1.0 | 1.05 |
| 300 | 0.75 | 0.9 | 0.95 | 1.0 | 1.05 | 1.2 |

* The characteristic periods of each model for the input peak acceleration of 25 gal are derived from Table 7.

From Figures 14–19 and Table 8, it is shown that the response spectrum characteristic period of each analysis model increases with the increasing input ground motion level. The response spectrum characteristic period of model 1 increases gradually from 0.50 s to 0.75 s; model 2 increases from 0.6 s to 0.9 s; model 3 increases from 0.7 s to 0.95 s; model 4 increases from 0.75 s to 1.0 s; model 5 increases from 0.8 s to 1.05 s; and model 6 increases from 0.85 s to 1.2 s. It can be observed that the characteristic period of the response spectrum increases roughly linearly with the increase in the input ground motion level.

In summary, the soft soil layer has a significant influence on the characteristic period of the site acceleration response spectrum. Compared with the Code [26], the site response spectrum characteristic period with the soft soil layer after the regulation is much larger than the value specified in the code. In the subsequent analysis, we will utilize the previously established 12 site models as analytical models to propose a method for adjusting the characteristic period of the acceleration response spectrum for a site with a soft soil layer. This method will be based on the impact of the thickness and burial depth of the soft soil layer on the characteristic period of the acceleration response spectrum. The aim is to provide a theoretical basis for determining the characteristic period of the seismic response spectrum for sites with soft soil layers.

Taking a unit-area soil column of height $h$ from a site with a soft soil layer, as shown in Figure 20a, the deformation of the column surface could be considered to be determined by the soft layer in the column. This assumes that the deformation of the upper and lower portions of the soil can be disregarded due to the low stiffness of the soft soil layer in comparison to its overlying and underlying soil layers. Based on this assumption, the soil column can be simplified to a spring–mass single degree of freedom system as depicted in Figure 20b, and the stiffness and mass of this system would be:

$$k = \frac{\rho_s \times v_s^2}{h_s} \tag{1}$$

$$m = h_s \rho_s + h_u \rho_u \tag{2}$$

where $\rho_s$, $v_s$, and $h_s$ are the density, shear wave velocity, and thickness of the soft soil layer, respectively. $\rho_u$ and $h_u$ are the thickness and density of the overlying soil layer, respectively.

It is not difficult to obtain the natural period of this spring–mass single degree of freedom system as follows:

$$T_0 = 2\pi \sqrt{m/k} \tag{3}$$

By combining Equations (1)–(3), it is easily seen that as the thickness of the soft soil layer or the overlying soil layer increases, the stiffness of the spring–mass single degree of freedom system decreases. Additionally, the mass of the system increases, resulting in an increase in the natural period of the system. This inevitably affects the spectral characteristics of the site seismic response, which in turn increases the characteristic period of the site seismic response.

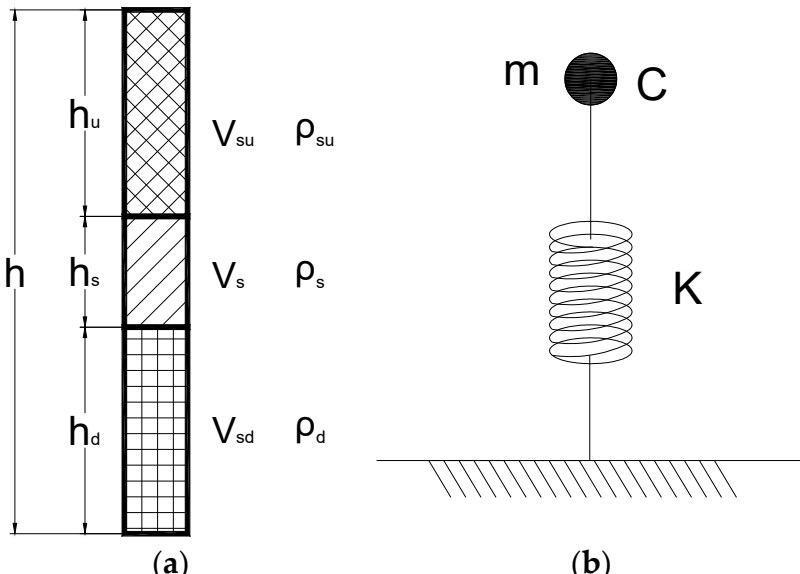

**Figure 20.** Analysis diagram: (**a**) Soil column; (**b**) spring–mass single degree of freedom system.

It is not difficult to explain the properties of the increase in the characteristic period of the site with the soft soil layer when the input intensity of ground motion increases. As the input ground motion intensity strengthens, several effects occur. Firstly, the strain level of the shallow surface soft soil layer increases. This leads to greater nonlinearity in the soft soil body. Secondly, the shear modulus ratio attenuates at a faster rate, resulting in a smaller ratio. Consequently, the shear modulus decreases, causing a decrease in the shear stiffness of the soft soil layer, This behavior is illustrated in Figure 20b for the spring–mass single degree of freedom system. As a result, the natural period of the system increases, subsequently leading to a greater characteristic period of the site with a soft soil layer. This qualitative interpretation also applies to the case where the difference between $V_{su}$, $V_{s}$, and $V_{sd}$ is not very large. Normally, the results of seismic response analysis of sites with soft soil layers indicate that the strain maximum occurs in the soft layer. Furthermore, the shear modulus ratio of the soft soil attenuates faster, which leads to the shear modulus ratio being smaller than that of the overlying and underlying soil layers. As a consequence, the difference in shear modulus between the three layers becomes larger, and this difference becomes more significant with increasing input ground motion intensity. Therefore, the aforementioned hypothesis is valid.

## 5. Discussion

To further illustrate the reasonableness of the three wave velocities when their differences are not significant, we made modifications to the original model 2 (see Table 9). Specifically, we replaced the silt layer with an overlying and underlying silty clay, resulting in the creation of two analytical models. These models are documented in Tables 10 and 11. Subsequently, we calculated and obtained the corresponding acceleration response spectra, which are displayed in Figures 21 and 22.

Figures 21 and 22 clearly demonstrate that there is a noticeable difference in the characteristic period of the response spectra when the wave velocity of the silt layer is not significantly different from that of up-and-down soil layers. This difference becomes more obvious with increasing input ground motion intensity.

**Table 9.** The original analytical model.

| No. | Rock-Soil | Soil Class | Depth at the Bottom of Layer (m) | Layer Thickness (m) | Shear Wave Velocity (m/s) | Density (t/m$^3$) |
|---|---|---|---|---|---|---|
| 1 | silty clay | 3 | 4.5 | 4.5 | 160 | 1.86 |
| 2 | silt | 1 | 9.5 | 5.0 | 112 | 1.58 |
| 3 | silty clay | 4 | 13.0 | 3.5 | 165 | 1.87 |
| 4 | silty clay | 5 | 17.0 | 4.0 | 199 | 1.88 |
| 5 | silty clay | 6 | 21.0 | 4.0 | 212 | 1.96 |
| 6 | silty clay | 7 | 24.0 | 3.0 | 242 | 1.98 |
| 7 | rounded gravel | 8 | 27.0 | 3.0 | 258 | 2.20 |
| 8 | fully weathered andesite | 8 | 30.0 | 3.0 | 393 | 2.25 |
| 9 | bedrock of model | 9 | | | 516 | 2.65 |

**Table 10.** The supplementary analysis model 1.

| No. | Rock-Soil | Soil Class | Depth at the Bottom of Layer (m) | Layer Thickness (m) | Shear Wave Velocity (m/s) | Density (t/m$^3$) |
|---|---|---|---|---|---|---|
| 1 | silty clay | 3 | 4.5 | 4.5 | 160 | 1.86 |
| 2 | silty clay | 3 | 9.5 | 5.0 | 160 | 1.86 |
| 3 | silty clay | 4 | 13.0 | 3.5 | 165 | 1.87 |
| 4 | silty clay | 5 | 17.0 | 4.0 | 199 | 1.88 |
| 5 | silty clay | 6 | 21.0 | 4.0 | 212 | 1.96 |
| 6 | silty clay | 7 | 24.0 | 3.0 | 242 | 1.98 |
| 7 | rounded gravel | 8 | 27.0 | 3.0 | 258 | 2.20 |
| 8 | fully weathered andesite | 8 | 30.0 | 3.0 | 393 | 2.25 |
| 9 | bedrock of model | 9 | | | 516 | 2.65 |

**Table 11.** The supplementary analysis model 2.

| No. | Rock-Soil | Soil Class | Depth at the Bottom of Layer (m) | Layer Thickness (m) | Shear Wave Velocity (m/s) | Density (t/m$^3$) |
|---|---|---|---|---|---|---|
| 1 | silty clay | 3 | 4.5 | 4.5 | 160 | 1.86 |
| 2 | silty clay | 4 | 9.5 | 5.0 | 165 | 1.87 |
| 3 | silty clay | 4 | 13.0 | 3.5 | 165 | 1.87 |
| 4 | silty clay | 5 | 17.0 | 4.0 | 199 | 1.88 |
| 5 | silty clay | 6 | 21.0 | 4.0 | 212 | 1.96 |
| 6 | silty clay | 7 | 24.0 | 3.0 | 242 | 1.98 |
| 7 | rounded gravel | 8 | 27.0 | 3.0 | 258 | 2.20 |
| 8 | fully weathered andesite | 8 | 30.0 | 3.0 | 393 | 2.25 |
| 9 | bedrock of model | 9 | | | 516 | 2.65 |

A qualitative explanation for the increase in the characteristic period of seismic response of the site with soft soil is given in the previous discussion. To further quantitatively characterize the impact of the thickness and burial depth of the soft soil layer on the site characteristic period, it is proposed to establish the influence law of the thickness and burial depth of the soft soil layer on the site characteristic period via the statistical regression method. This method was based on the analysis data of the above models and the influence law of the input ground motion intensity on the site characteristic period of the soft soil layer.

$$T_g = a \times T_0 + T'_g + b \times T \tag{4}$$

Based on the qualitative analysis presented above, it is clear that the characteristic period of a site with a soft soil layer is influenced by the site's dominant period. Consequently, it is assumed that the characteristic period ($T_g$) of a site containing a soft soil layer is determined by the natural period ($T_0$) of the soft soil layer, the characteristic period of the input ground motion ($T'_g$), and the dominant period of the underlying soil ($T$).

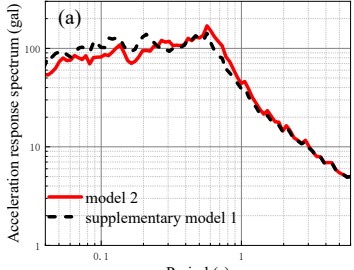
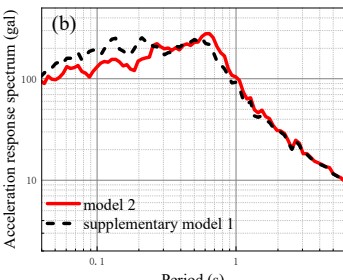
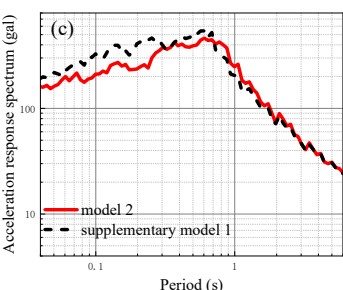

**Figure 21.** The site-related acceleration response spectra of supplementary model 1 and analysis model 2 under different input ground motions: (**a**) Input seismic peak acceleration 25 gal; (**b**) input seismic peak acceleration 50 gal; (**c**) input seismic peak acceleration 100 gal.

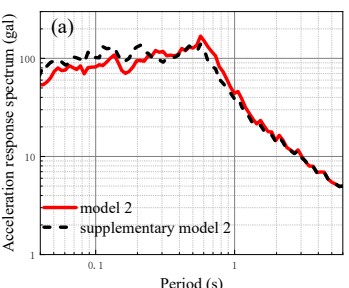
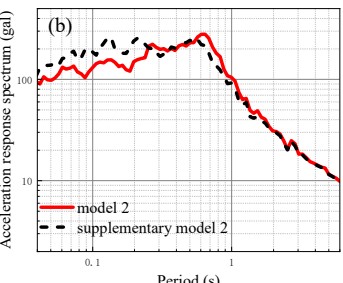
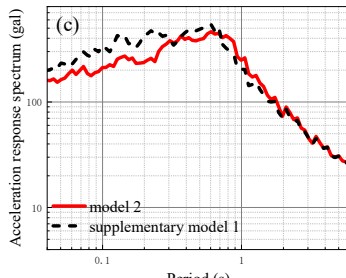

**Figure 22.** The site-related acceleration response spectra of supplementary model 2 and analysis model 2 under different input ground motions: (**a**) Input seismic peak acceleration 25 gal; (**b**) input seismic peak acceleration 50 gal; (**c**) input seismic peak acceleration 100 gal.

Where $T = \frac{4h_d}{v_{sd}}$, hypothetically, and $v_{sd}$ is the equivalent shear wave velocity of the soil underlying the soft layer. $h_d$ is the thickness of the soil underlying the soft layer; *a* and *b* are the regression coefficients.

From Equation (4), it is known that the site characteristic period takes into account the influence of the input ground motion spectrum characteristics, the overlying and underlying soils, and the soft soil layer. Based on the data in Tables 7 and 8, the regression coefficients a and b in Equation (4) are available by the least squares method, as listed in Table 12, and the regression results are presented in Figure 23.

**Table 12.** Summary of fitting results for a characteristic period of the site response spectrum under different input ground motion levels.

| Models 1–6 | | | | Models 7–12 | | | |
|---|---|---|---|---|---|---|---|
| Input Ground Acceleration (gal) | a | b | $R^2$ | Input Ground Acceleration (gal) | a | b | $R^2$ |
| 25 | 0.838 | −0.066 | 0.9998 | 25 | 0.972 | −0.297 | 0.99912 |
| 50 | 0.941 | −0.166 | 0.99905 | 50 | 1.050 | −0.410 | 0.99964 |
| 100 | 1.231 | −0.270 | 0.99972 | 100 | 1.246 | −0.223 | 0.99938 |

When the variables are analyzed by linear regression in statistics, the parameters are estimated by the least squares method. $R^2$ is the ratio of the sum of squares of regression to the sum of squares of total deviation, which represents the proportion that can be explained by the sum of squares of regression. The larger the ratio, the more accurate the model and the more significant the regression effect. $R^2$ ranges from 0 to 1, where a value of 1 indicates a perfect fit of the model to the data. It is generally believed that the goodness of fit of models exceeding 0.8 is higher. The statistical relationship $R^2$ between the characteristic period of the site with a soft soil layer and the burial depth of the soft layer for the same thickness of soft soil layer at different input levels is about 1. This indicates that the

correlation between the characteristic period of the site response spectrum and the burial depth of the soft soil layer is relatively good at different input ground motion levels. The variation curves of the characteristic period of the site response spectrum with the soft soil layer buried at different input ground motion levels for models 1–6 are approximately straight lines. Additionally, the variation curves of the characteristic period of the site response spectrum with the soft soil layer buried at different input ground motion levels for models 7–12 have greater curvature than those for models 1–6. Furthermore, it is observed that the contribution value of the natural period of the soft soil layer to the characteristic period increases as the thickness of the soft soil layer increases.

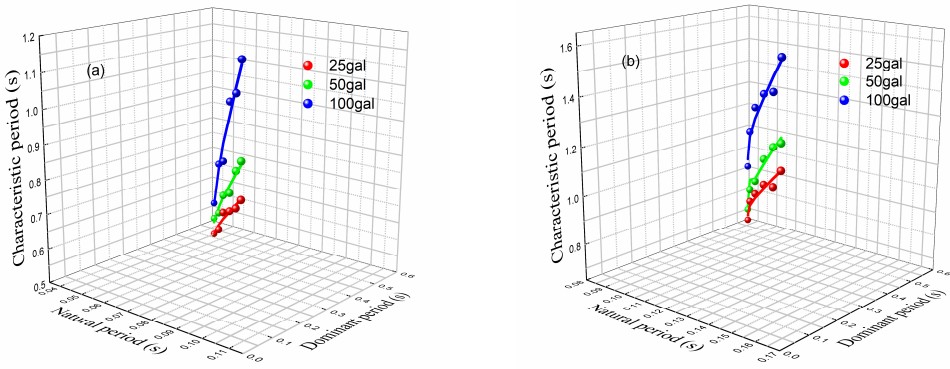

**Figure 23.** Relationship between the characteristic period of site response spectrum and thickness and buried depth of soft soil layer under different input ground motion levels: (**a**) Models 1–6; (**b**) Models 7–12.

In addition, the variation trend of the characteristic period $T_g$ of the response spectrum, in relation to the input peak ground motion acceleration A, is obtained at the site with a specific thickness and depth of burial of soft soil layers. This information is derived from Table 8 and depicted by the scatter points in Figure 24. It is easy to see from Figure 24 that the characteristic period $T_g$ of the site acceleration response spectra of different models is similar with an increasing peak acceleration of input ground motion. It is approximately linear with the input peak acceleration $A$. Therefore, the equation $T_g = \alpha + \beta \times A$ is proposed to fit the variation law of the characteristic period $T_g$ of the site acceleration response spectrum with the input peak acceleration $A$. The regression coefficients $\alpha$ and $\beta$ in the equation are obtained by linear regression based on the least squares method, as listed in Table 13, and the regression results are demonstrated as straight lines in Figure 24.

The regression analysis results, indicated by the high $R^2$ values close to 1, indicate a strong correlation between the characteristic period of the site response spectrum and various input peak ground motion accelerations. Moreover, the characteristic period of the response spectrum exhibits an approximately linear increase with the input peak acceleration for the site, which includes a specific thickness and burial depth of a soft soil layer. This finding suggests that as the input peak ground motion acceleration increases, the characteristic period of the site response spectrum also lengthens.

By considering the influence of soil structure on the characteristic period of the site seismic acceleration response spectrum, we propose a straightforward and physically meaningful adjustment formula. This formula considers not only the effects of soft soil layer thickness and burial depth but also the influence of different input ground motion levels. The proposed formula provides a clear and concise representation of these factors. The seismic response analysis conducted using a representative soil structure site model demonstrates the effectiveness of the adjustment method. The method accurately reflects the influence of the burial depth of the soft soil layer within the profile on the characteristic period of the site seismic acceleration response spectrum. Furthermore, the estimated value obtained from the adjustment method closely approximates the characteristic period

of the response spectrum observed at the actual engineering site. This indicates that the adjustment method provides reliable and realistic results.

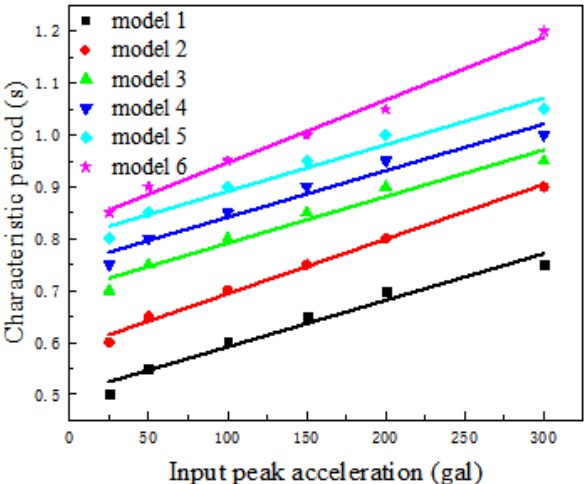

**Figure 24.** Variation of characteristic periods of seismic response spectra of different models with different input ground motion levels.

**Table 13.** Fitting results of characteristic periods of models 1–6 at different input levels.

| Analytical Model | $\alpha$ | $\beta$ | $R^2$ |
|---|---|---|---|
| 1 | 0.5015 | 0.0009 | 0.9624 |
| 2 | 0.5884 | 0.0010 | 0.9937 |
| 3 | 0.7015 | 0.0009 | 0.9624 |
| 4 | 0.7515 | 0.0009 | 0.9624 |
| 5 | 0.8015 | 0.0009 | 0.9624 |
| 6 | 0.8254 | 0.0012 | 0.9953 |

## 6. Conclusions

Based on the established 12 site models with soft soil layers, the one-dimensional equivalent linearization site seismic response analysis is carried out under different input ground motion levels. The effects on the soft soil layer thickness, buried depth, and input ground motion intensity of the on-site seismic response, as well as the characteristic period of the site acceleration response spectrum, have been discussed. In light of these findings, a method for adjusting the characteristic period of the site acceleration response spectrum with soft soil layers is proposed. The main conclusions obtained are as follows:

1.  Under the same input ground motion level, the burial depth of the soft soil layer influences the peak ground acceleration. Specifically, as the burial depth increases, the peak ground acceleration decreases, and the rate of attenuation is faster near the surface. Conversely, as the burial depth increases further, the rate of attenuation becomes slower. Under different input ground motion levels, higher input peak accelerations result in faster attenuation of the peak ground acceleration. In other words, as the input peak acceleration increases, the rate of attenuation becomes more rapid. Similarly, the thickness of the soft soil layer impacts the peak ground acceleration. A thicker soft soil layer leads to a smaller peak ground acceleration, and the attenuation is faster near the shallow surface. The difference becomes more prominent as the input ground motion peak acceleration increases.

2.  At the same input ground motion, the dynamic amplification coefficient of the peak surface acceleration decreases with an increase in the burial depth of the soft soil layer. Additionally, the attenuation of the dynamic amplification coefficient is faster near the surface and slower as the burial depth increases. Under different input ground motion

levels, higher input peak acceleration results in a faster attenuation of the dynamic amplification coefficient. In other words, as the input peak acceleration increases, the dynamic amplification coefficient decreases at a faster rate. Likewise, the thickness of different soft soil layers has an effect on peak surface acceleration. A thicker soft soil layer corresponds to a smaller dynamic amplification coefficient, and the attenuation is faster near the shallow surface. The difference becomes more obvious as the input ground motion peak acceleration increases.

3. The characteristic period of the site seismic acceleration response spectrum progressively increases with the increase in burial depth or thickness of the weak soil layer. It also increases with the increase in the input ground motion peak. Subsequently, the influence characteristics of soft soil layer thickness, buried depth, and input ground motion intensity on the characteristic period of the site seismic acceleration response spectrum are analyzed. Furthermore, a method for adjusting the characteristic period of the site seismic acceleration response spectrum with the soft soil layer is proposed.

**Author Contributions:** Conceptualization, Z.Z. and Z.C.; methodology, Y.L.; software, B.H.; validation, Y.L., B.H. and Y.H.; formal analysis, Y.L.; investigation, Z.B.; resources, C.P.; data curation, Y.L.; writing—original draft preparation, Y.L.; writing—review and editing, Y.L. and Z.C.; visualization, B.H.; supervision, Z.B., Y.H. and C.P.; project administration, Z.Z.; funding acquisition, Z.Z. All authors have read and agreed to the published version of the manuscript.

**Funding:** This research was funded by the [National Natural Science Foundation of China], grant numbers [U2039208] and [U1839202].

**Institutional Review Board Statement:** Not applicable.

**Informed Consent Statement:** Not applicable.

**Data Availability Statement:** Not applicable.

**Conflicts of Interest:** The authors declare no conflict of interest.

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
