# Peer review of "Discussion on Adjustment Method of the Characteristic Period of Site Response Spectrum with Soft Soil Layer"

_sustainability, doi:10.3390/su15118837_

Round 1

Reviewer 1 Report

The manuscript investigates Discussion on adjustment method of the characteristic period 2 of site response spectrum with soft soil layer.

 The results show that the characteristic period of the response spectrum gradually increases with the increase of buried depth or thickness of the soft soil layer, and the characteristic period of the response spectrum also increases with the increase  of the input ground motion peak. Furthermore, according to the influence characteristics of soft soil thickness, buried depth, and input ground motion intensity on the characteristic period of the site 16 acceleration response spectrum, a method for adjusting the characteristic period of the site acceleration response spectrum with soft soil layer is put forward..

1.Scientific writing: The scientific writing of the manuscript requires significant revision. I would like to suggest the manuscript to be professionally proofread and edited. Moreover, the authors may pay attention to some aspect of the conventional research writing, especially the connection between the sentences, the components/structure of the key parts (Abstract, Introduction, body, Conclusion). I suggest the authors reading the following references to learn more about the scientific writing and modify the paper accordingly.

2. Abstract: The abstract is the summary of a manuscript. It should include the problems, methodology to solve the problems and main points of conclusions. Suggest to control the total words of abstract within 150 words.

3. Introduction: It must contain background, in which problems should be proposed; the present state of the art on the research of the problem, the gap of the present research and the topics to investigate, and objectives of the present research. Length of the Introduction should be 1/8~1/10 (one-eighth to one-tenth) of the paper.

4.Results and Discussion: In this section, authors should tell all the story of the research; what was observed and what was confirmed? This section must show coherence in the results and discussion procedure and should be simple to understand. They should be referred to statistical analysis and describe the trends observed and explain the significance of the project results on large scale understanding.  The results should be a critical analysis of the data collected in the field.

5.Conclusions: The conclusions should be summarizing the innovative points of the new findings after research. The length of conclusions should be controlled within 1/15~1/20, for the manuscript.

Author Response

We sincerely thank you for taking your precious time to provide constructive comments and valuable suggestions, which greatly raised the quality of the manuscript and enabled us to improve it. Each of the suggested revisions and comments was carefully incorporated and considered. Please see the attachment for the point-by-point response.

Reviewer 2 Report

In the context of earthquake engineering, the authors study samples of soft sites containing a silt layer and analyze the influence of the soft soil layer on the seismic response of the site by proposing a fitting method for the characteristic period of the response spectrum. .

In addition, they provide a theoretical basis for the determination of the characteristic period of the seismic response spectrum of the site with soft soil layer.

After carefully reading the manuscript, I consider that it can be published with some details.

1. The gal unit for acceleration is not very common, so authors must specify their relationship in the international MKS system.

2. Line 107: The phrase “For analyzing the seismic response of the site with soft soil layer”, is repeated "in order to analyze the seismic response of the site with soft soil layer".

3. To avoid confusion, they must specify that the term models refer to samples.

4. the authors do not describe how the parameter shear wave velocity (m/s) was calculated, in Tables 1 and 2,

5. The authors do not describe how the shear strain parameter was calculated in Table 3.

The discussion and conclusions seem well structured to me.

Author Response

(The authors gave the same response as above.)

Reviewer 3 Report

Major concerns:

-Add the doi identifications of all references to simplify the consultation;

- Check others references that introduce the theme explained in this study.

Minor concerns:

- in pag. 2 compare 'Tian et al.' but in the references is 'Tian, 2013';

-in line 81 replace 'determning' with 'determining';

- in Figure 1 there is a double panel 'c' and is absent the panel 'd';

- please reformulate the phrase among lines 107-112;

- please improve the explication (probably could be necessary with others tables) the different models of profiles and mechanical parameters between table 1 and table 2;

- remove the anomalous lines in the top line of tables 4, 5, 7, 8;

- improve the methods used in this study;

- line 175 replace ';' with '.'

- add to the references the Code for Seismic design of buildings;

- add the numbers of different panels and explain in the caption of figures from 8 to 13;

- correct the commas on line 392;

- figures 21, 22, 23 should be represented with different colours to better represent the results obtained.

Author Response

(The authors gave the same response as above.)

Round 2

Reviewer 1 Report

Accept in present form

Reviewer 3 Report

Thanks for the answer.

Congratulation!